# Monoclonal Antibodies for Bacterial Pathogens: Mechanisms of Action and Engineering Approaches for Enhanced Effector Functions

**DOI:** 10.3390/biomedicines10092126

**Published:** 2022-08-30

**Authors:** Fabiola Vacca, Claudia Sala, Rino Rappuoli

**Affiliations:** 1Department of Biotechnology, Chemistry and Pharmacy, University of Siena, 53100 Siena, Italy; 2Monoclonal Antibody Discovery Laboratory, Fondazione Toscana Life Sciences, 53100 Siena, Italy; 3Faculty of Medicine, Imperial College, London SW7 2BX, UK

**Keywords:** monoclonal antibodies, infectious diseases, Fc engineering, bacteria, antibiotic resistance

## Abstract

Monoclonal antibody (mAb) therapy has opened a new era in the pharmaceutical field, finding application in various areas of research, from cancer to infectious diseases. The IgG isoform is the most used therapeutic, given its long half-life, high serum abundance, and most importantly, the presence of the Fc domain, which can be easily engineered. In the infectious diseases field, there has been a rising interest in mAbs research to counteract the emerging crisis of antibiotic resistance in bacteria. Various pathogens are acquiring resistance mechanisms, inhibiting any chance of success of antibiotics, and thus may become critically untreatable in the near future. Therefore, mAbs represent a new treatment option which may complement or even replace antibiotics. However, very few antibacterial mAbs have succeeded clinical trials, and until now, only three mAbs have been approved by the FDA. These failures highlight the need of improving the efficacy of mAb therapeutic activity, which can also be achieved with Fc engineering. In the first part of this review, we will describe the mechanisms of action of mAbs against bacteria, while in the second part, we will discuss the recent advances in antibody engineering to increase efficacy of pre-existing anti-bacterial mAbs.

## 1. Introduction

Monoclonal antibodies (mAbs) represent a valid treatment option for various diseases [1] and are one the most promising classes of biological drugs on the pharmaceutical market [2]. Antibody therapy was first introduced with the use of immune sera-derived immunoglobulins, which contain different types of antibodies. As mAbs are chemically defined reagents with low lot-to-lot variability, they eventually replaced polyclonal preparations [3]. They can be generated in the laboratory with different approaches, such as hybridoma technology, molecular cloning, and recombinant expression, that can yield humanized or fully human antibodies [4]. Moreover, via single B cell screening technologies, it is now possible to generate mAbs from single B cells obtained from immunized animals or infected individuals [5].

The first monoclonal antibody to be approved by the United States Food and Drug Administration (FDA) for therapy was muromonab-CD3 (Orthoclone OKT3^®^) [6]. After approval in 1985, it was used to treat organ-transplant-associated rejections [6]. Since then, as of July 2021, 100 mAbs have obtained approval by the FDA [7]. The progressive increasing interest is partially due to the fact that mAbs are generally well-tolerated, highly specific, and with low off-target effects [8,9]. Their production is feasible, as pharmaceutical companies have adopted well-designed platform processes to robustly manufacture and develop mAbs [10]. Additionally, due to the recent advances in bioinformatic tools and studies in genomics and proteomics, many new potential mAb targets have been discovered to modulate disease, allowing the study of diseases and pathogens at the molecular level [11]. Among the five isotypes of antibodies (IgG, IgA, IgM, IgD, and IgE), IgG is the main class in the list used as therapeutic. It has an extremely long half-life, high serum abundance, and it is suitable for protein engineering [12]. IgG is composed of two identical fragment antigen-binding (Fab) domains, which mediate binding to the target, and one fragment-crystallizable (Fc) domain. Fab and antigen engagement is crucial for ensuring a specific response, but the constant domains of the heavy chain are equally important. They are responsible for the recognition of different receptors (FcgR, FcRn, complement), act on the effector function of the antibody, and form an essential link between innate and adaptive immunity [13,14]. In fact, when IgG interacts with FcgRs, the outcome of the interaction depends on the expression pattern of the receptors on effector cells and on the affinity of the Fc domain for the specific receptor [15]. The Fc domain has been the central hub for mAb engineering, meant to improve effectiveness, eliminate side-effects, and enhance safety and half-life. Through mutation of selected residues within the Fc domain, the Fc effector function can be modified and interaction with its receptors modulated [16,17]. Therapeutic mAbs find application in different areas of research, including cancer and autoimmune and metabolic diseases [18]. Major research efforts have also been expended in the infectious diseases field. Most importantly, considering the emerging issue of antibiotic resistance in many bacterial pathogens, mAbs have gained progressively more attention as an alternative anti-bacterial therapeutic approach, due to their role in mediating host defense against bacteria [19]. Continuous globalization and unrestrained antibiotic usage predict a dramatic rise of antibiotic-resistant strains, meaning that soon, some strains may become impossible to eliminate. Therefore, antibody-based intervention may progressively become essential to overcome antibiotic resistance in difficult-to-treat pathogens [20]. However, despite current efforts, only three anti-bacterial mAbs have been approved by the FDA to date: raxibacumab, obiltoxaximab, and bezlotoxumab. Raxibacumab is a human recombinant IgG1 mAb developed against the protective antigen toxin (PA) of *Bacillus anthracis*. It received approval by the FDA in 2012 for the treatment of adult and pediatric patients with inhalational anthrax in combination with appropriate antibiotic drugs [21,22]. Its safety and efficacy, at the approved dose of 40 mg/kg, was assessed through clinical studies involving more than 300 healthy adults [21]. In 2016, obiltoxaximab, a chimeric IgG1 against PA, obtained FDA approval for the prevention and treatment of inhalational anthrax [23,24], after its efficacy and safety in animal models and in healthy human volunteers had been observed [24]. Lastly, IgG1 Bezlotoxumab, the first mAb drug targeting *Clostridium difficile* toxin B, has been approved by the FDA in 2016 for recurrent infections [25]. It can reduce the recurrence of *C. difficile* infections (CDI) in patients of 18 years of age or older who receive antibacterial drug treatment for CDI and are at a high risk for CDI recurrence, but it is not indicated for the treatment of CDI and is not considered an antibacterial drug. Its safety and efficacy were investigated in two Phase 3, randomized, double-blind, placebo-controlled studies. The main adverse effect, heart failure, has been observed in patients with underlying congestive heart failure (CHF) [26].

Generally, one of the main reasons for clinical trial failure of anti-bacterial mAbs is the difficulty in translating results from in vivo tests to clinical trials. Animal models have different genetic and immunological backgrounds compared to humans, thus data obtained with animals do not necessarily mirror clinical trials in humans [4]. In order to understand and predict any adverse toxic effects in humans, it is crucial to use animal species which are pharmacologically relevant. Given the high specificity thar characterizes human mAbs, often non-human primates, rather than mice or rabbits, are the only pharmacologically relevant species [27]. For example, in the case of sexually transmitted infections, whose main site of infection is the uro-genital tract, various issues are faced. The anatomical site of infection might not resemble that of humans, including human-specific receptors for bacterial adherence and invasion, and animals have a different estrous cycle and gestational period. For example, *Chlamydia trachomatis,* a sexually transmitted bacterial pathogen, lacks mouse models and researchers mostly rely on guinea pigs [28]. Nevertheless, mouse models which mimic the human infection have been established and used for mAb testing. In case of *N. gonorrhoeae*, which is strictly adapted to humans, an experimental model of genital tract infection has been developed in estradiol-treated female mice to study mechanisms of infection and to test mAbs [29]. In fact, Gulati et al. managed to test a mAb which recognized a gonococcal lipooligosaccharide in a mouse vaginal colonization model and concluded that mouse models can be appropriate for evaluating Fc-mediated effects of human IgG1 mAbs. Therefore, despite anatomical differences, mouse models can be a tool to characterize mAbs functionality in an in vivo condition [30].

As bacterial pathogens are becoming extremely difficult to target, it is essential to use efficient therapeutic approaches to inhibit their virulence mechanisms. In this review, we summarize the mechanisms through which mAbs act against bacterial pathogens. In addition, since Fc engineering provides new opportunities to exploit and improve antibody therapeutics, we will discuss the recent advances in antibody modification and application to increase efficacy of pre-existing anti-bacterial mAbs.

## 2. Mechanisms of Action of mAbs against Pathogenic Bacteria

Bacterial pathogenesis involves different components and virulence mechanisms, and mAbs have proved to be able to act throughout multiple pathogenesis steps (Figure 1).

It is possible for mAbs to bind to released molecules, such as toxins or quorum-sensing signaling molecules [31], cell-surface components (proteins and exopolysaccharides), and polysaccharide structure of capsulated bacteria [32]. Targetable structures include the lipopolysaccharide (LPS), the capsular polysaccharide, and the outer membrane vesicles.

LPS is the main constituent of Gram-negative bacteria, and it consists of lipid A, core oligosaccharide, and O-specific oligosaccharide chain. Given its ability to specifically activate Toll-like receptor 4 and induce inflammation, LPS is a highly immunogenic molecule [33]. For example, LPS of *Legionella pneumophila* was identified as the target structure against which antibodies were generated and evaluated [34]. Cohen et al., who worked on anti–*K. pneumoniae* LPS-O-antigen mAbs, compared the efficacy of two mAbs, which were both capable of enhancing neutrophil-mediated opsono-phagocytic killing, but differed in their neutralization activity. They showed that LPS neutralization significantly reduced mAb protection in mouse infection models, while opsono-phagocytic killing was confirmed as the main mechanism of action. Therefore, when LPS is taken into consideration as a potential mAbs target, it is crucial to understand if mAbs exert their main protective activity in synergy with the host immune system [35].

The capsular polysaccharide is composed of tightly packed repeating polysaccharide units around the bacterial cell wall. It is associated with protection from host immune responses and resistance to antimicrobial compounds, which is the reason for why mAbs that target this structure would be of particular relevance [36]. In the case of *Streptococcus pneumoniae*, it is thought that antibodies against the capsular polysaccharide could provide a high level of protection [37].

Outer membrane vesicles are nanostructures released by Gram-negative bacteria and derive from the outer membrane. They constitute a useful tool for mAb discovery as they are heterogeneous in size and composition, and are composed of proteins, lipids, and glycans and encapsulate soluble periplasmic content [38]. In the case of *Neisseria meningitidis*, outer membrane vesicles have been used as a vaccine platform because they are nonreplicating, immunogenic mimics of their parental bacteria and can be exploited to assess mAbs binding without the interference of cytoplasmic proteins [39]

Among the main mechanisms of action, mAbs may act through neutralization and inhibition of adhesion, which are mainly Fab mediated, and through antibody-dependent cellular phagocytosis (ADCP), antibody-dependent cellular cytotoxicity (ADCC), and complement-dependent cytotoxicity (CDC), which depend on the Fc. Neutralization takes place mainly due to the ability of the variable region of the Fab portion to bind toxins and interfere with the binding of toxins to targets. All the FDA-approved mAbs target endotoxins by neutralization. Successful infection by bacterial pathogens requires adhesion to host cells, using adhesins on the surface to bind specific receptors on the host cells, followed by colonization and invasion [40]. In the case of *Bordetella pertussis,* when it first interacts with epithelial cells, an enhanced bacterial growth is observed [41]. Cells have been reported to release biological factors which support growth, and through interactions with bacterial adhesins, induce signaling for bacterial replication. Antibodies interfering with this process can thus inhibit the infection and avoid the formation of an environment favoring bacterial growth. It has been shown that mAbs, which target the filamentous hemagglutinin (FHA) and fimbriae 2 and 3 (FIM) of *B. pertussis*, were capable of inhibiting adhesion of the bacteria onto epithelial cells in vitro [41,42].

Also, mAbs can opsonize pathogens to facilitate phagocytosis in phagocytic cells such as monocytes, macrophages, or neutrophils. Opsono-phagocytic activity can be crucial, as it is considered a major predictor of antibody protective efficacy [32]. However, some bacteria have developed intracellular survival mechanisms, escaping from the lysosome or inhibiting its formation [43]. Nevertheless, mAbs may also enhance phagosome maturation, restricting the survival of the bacterium in macrophages [44,45] or promoting neutrophil extracellular traps (NETs) release. Two anti-capsular polysaccharide mAbs developed against invasive infections caused by hypervirulent (hv) *Klebsiella pneumoniae* were shown to work mainly through FcR-mediated phagocytosis and enhancement of NETs release. Activity was also confirmed in vivo, as they exhibited protective efficacy in multiple mouse models. In particular, in colonized mice, they were able to decrease dissemination of the hv *K. pneumoniae* from the gut to other organs [32].

Besides ADCP, antibodies can also induce activation of the complement cascade. This contributes to pathogen elimination either directly, by means of CDC, or indirectly, through phagocytic clearance of complement-coated targets and the induction of an inflammatory response [14]. Other possible inhibitory roles of mAbs investigated in vitro are the inhibition of biofilm formation and quorum-sensing system. Biofilms constitute bacterial communities that adhere to abiotic surfaces using a self-made extracellular matrix composed of proteins, polysaccharides, and extracellular DNA. Due to their biophysical architecture and quiescent metabolism [46], biofilms can protect bacteria from host immunity [47] and antibiotic therapy [48], thus playing a major role in the survival of bacterial pathogens. Biofilm formation is mediated by membrane-bound protein and carbohydrate factors, which act as adhesins that mediate cellular attachment to abiotic surfaces [49]. Antibodies acting against those molecules could disrupt cell-to-cell and cell–surface interactions, thereby interfering with biofilm formation [50]. Sun. et al. identified three mAbs against cell-wall-bound accumulation-associated protein (AAP) in *Staphylococcus epidermidis* that were capable of inhibiting biofilm formation on abiotic surfaces. When used singly, these mAbs inhibited *S. epidermidis* biofilm formation by up to 66%, while two out of three mAb combinations inhibited biofilm formation by 79 and 87% [50]. Another group identified several human mAbs capable of detecting *Staphylococcus aureus* biofilms in vitro and in vivo. They grouped mAbs into two classes: one that uniquely binds *S. aureus* in biofilm state and one that recognizes *S. aureus* in both biofilm and in planktonic state (i.e., free-floating bacteria). Using mAbs that target bacteria in both states offers the possibility to target *S. aureus* in vivo throughout the entire infection cycle [51]. If mAbs can work in synergy with antibiotics, when biofilm is disrupted, dispersed bacteria would reverse to a more drug-sensitive planktonic state and thus be susceptible to administered antibiotics [52]. Ibáñez de Aldecoa et al. demonstrated that an mAb against a component of the biofilm matrix of *S. aureus* can disrupt biofilm. Biofilms contain extracellular DNA (eDNA), which can form a three-dimensional mesh-like structure that defends bacteria from immune cells but still allows the exchange of both nutrients and waste [53]. To be stabilized, eDNA must be bound by bacterial proteins. A human mAb which binds to a family of eDNA stabilizing proteins has shown efficacy in vitro, disrupting established biofilms of *S. aureus*, but also *Pseudomonas aeruginosa* and *Acinetobacter baumannii*. Furthermore, it has proved to increase antibiotic susceptibility in different mouse models [52].

Lastly, interference with quorum sensing has been explored as an approach for inhibiting biofilm formation [52]. In response to fluctuations in cell-population density, bacteria communicate through the release of chemical signaling molecules (autoinducers) to regulate gene expression and influence virulence activities [54]. A type of signaling molecule is represented by the *N*-acyl homoserine lactones (AHLs), which have highly conserved components and are released extracellularly [55]. *P. aeruginosa* uses an AHL molecule called 3-oxo-C12-HSL which plays a regulatory role, influencing virulence factor expression and biofilm formation. In addition, it can be quite toxic to eukaryotic cells, inducing apoptosis in macrophages. An antibody called RS2-1G9 has been found to target the quorum-sensing system of *P. aeruginosa*. Its ‘quorum-quenching’ activity is given by its capacity to bind the QS molecule, protecting cells from apoptosis and inhibiting the activation of cellular stress kinase pathways [31].

## 3. Factors Influencing mAb Effector Functions

Understanding mAb mechanisms of action is crucial. However, it is equally important to understand what facilitates or impairs these mechanisms to develop more effective therapeutic mAbs. Despite the efforts aimed at the production of therapeutic mAbs against bacterial pathogens, many candidates were not successful in mitigating the severity of bacterial infections in preclinical and clinical trials. First, one candidate mAb, which binds a single antigen, may not provide adequate protection against bacteria. Crucial virulence factors are often expressed at low levels, resulting in low abundance of the antigen on the membrane. Therefore, mAb binding may not be sufficient to elicit a response [56]. Furthermore, one single mAb may not be able to cover a broad range of strains and serotypes of the same pathogen [56] and, as bacteria have many targets on their surface, targeting one molecule may not be enough. This issue could be overcome by the co-administration of mAb combinations or by using recombinant polyclonal antibodies. For example, for *S. aureus*, Tkaczyk et al. showed that combining mAbs that target multiple virulence factors gives better results compared to each single mAb, providing improved efficacy and broader strain coverage. They demonstrated that the combination of one mAb against clumping factor A (ClfA), an important virulence factor facilitating *S. aureus* bloodstream infections, and one mAb against cytolytic pore-forming alpha-toxin, neutralizes multiple virulence mechanisms and targets bacteria for opsono-phagocytic killing [57]. However, mAb combinations are more prone to face greater clinical development issues (the need of multiple bioprocesses and to produce and clinically test each individual mAb), which is why it is generally more practicable to manufacture a single molecule [56,57].

Additionally, the nature of the bound epitope can influence the effector mechanism of the mAb [58]. In fact, it seems that different mAbs that bind to the same antigen can trigger different effector mechanisms [59]. To explain this, it has been hypothesized that the distance between an epitope and the target cell membrane can be crucial, as the position of the bound mAb with respect to the cell surface can impact Fc-mediated mechanisms [60]. In case of CDC, given the short half-life of the active components of the cascade, if activated far away from the target cell membrane, the activated complement components have a reduced chance of stabilizing on the cell surface. Therefore, the greater the distance from the cell surface, the lower the efficiency in the establishment of the whole cascade and of the membrane attack complex (MAC) [58].

## 4. mAb Fc Engineering

The interaction between the Fab portion of the antibody and the antigen is crucial for the protective activity of the antibody. However, to fully exploit the potential of the antibody, the Fab-mediated recognition must be coupled with Fc effector activity. It has been demonstrated that the Fc can tolerate different mutations, each of which can drive different effector activities (Table 1) [61]. For example, Fc engineering can enhance phagocytic activity of mAbs in presence of phagocytic cells in vitro [62] and improve cell-based complement dependent cytoxicity and binding affinity to C1q [63].

### 4.1. Fc Engineering for Enhanced Receptor Engagement

Neutralizing mAb activity apparently relies on the Fab specificity for the antigen. However, there are doubts as to whether neutralization results only from the inhibition of binding of toxins to their targets. Bournazos et al. have shown that even in neutralizing antibodies, IgG interaction with FcγRs is crucial to ensure optimal efficacy, especially in vivo [15]. By manipulating the Fc to enhance affinity for human FcγR, it is possible to enhance the toxin-neutralizing activity of an anti-anthrax chimeric mAb. To exert its toxic activity and bind cell-surface receptors, PA must first undergo proteolytic cleavage [65]. An increased endocytic uptake of mAb-opsonized PA, resulting from increased engagement of mAb on activating FcγRs on effector cells, could prevent this cleavage. Overall, mutagenesis of the Fc domain resulted in more effective neutralizing antibodies, without influencing their binding affinity or specificity [15]

One approach to potentiate the complement-dependent killing is to exploit the capacity of IgG antibodies to organize into ordered hexamers. De Jong et al. demonstrated that by inserting specific mutations in the IgG1 backbone, antibodies rearrange into hexamers after binding to their antigen. Specifically, mutations E345K and E430G confer a stronger ability to induce complement-dependent cytotoxic activity of cell lines and chronic lymphocytic leukemia. An antibody targeting CD20 that was initially ineffective in activating complement displayed enhanced CDC- and antibody-dependent cellular cytotoxicity when the Fc portion was mutagenized to promote hexamerization [66]

Gulati et al. engineered a bactericidal mAb against *Neisseria gonorrhoeae* with the E430G Fc modification, enhancing hexamerization following target binding and increasing complement activation. The mAb engaged greatly with C1q, induced higher C4 and C3 deposition compared to wild-type mAb, which translated into increased bactericidal activity in vitro and, consequently, enhanced efficacy in vivo [30]. IgGs generated against the cell-wall-component teichoic acid of *S. aureus* were also modified with the hexamer-enhancing mutation E430G [67]. As Gram-positive bacteria are protected from MAC-dependent lysis by their thick peptidoglycan layer [68], the antibody was evaluated for its ability to enhance complement-dependent phagocytosis of the bacterium. Zwarthoff et al. first observed that the mutated mAb enhanced C3b deposition and improved the phagocytosis of bacteria by human neutrophils in serum. Most importantly, the E430G mutation did not affect the IgG-dependent phagocytosis in the absence of complement [67]. Another approach to improve the effector activity of the antibody relies on adding multiple copies of the Fc portion. This modification, which acts on protein folding, influences the functionality of the mAb. Wang et al. speculated that by increasing the number of copies of Fc up to three in an original anti- *K. pneumoniae* IgG1 antibody, ADCC and ADCP activities could be significantly improved. In vitro, these modifications enhanced opsonophagocytic killing activity, while in vivo they ameliorated the protective efficacy in a mouse infection model, without any adverse effects. It seems that the mechanism of protection behind this derives from the incremental interaction with FcgR, FcRn, and C1q through the avidity effect. However, one drawback of this approach is that this form undergoes faster clearance compared with its IgG1 counterpart [69].

### 4.2. Fc Engineering for Increasing Valency

Rather than using polyclonal Abs, a single mAb can be engineered to bispecificity, that is, to bind to two different antigens or two epitopes on the same antigen [70,71]. Bispecific mAbs tend to be classified as “IgG- like”, which contain an Fc region and resemble conventional antibodies and “non-IgG-like”, which lack the Fc region [58]. There are various platforms which can be used to produce bispecifc mAbs. For example, through Fc engineering, it is possible to fuse two antibodies into one by promoting heterodimer formation of two complementary CH3 domains on the Fc portion. One domain of the first Fc chain is modified to show a strong preference for pairing with the domain of the second Fc chain, rather than pairing with themselves [59].

Di Giandomenico et al. developed a bispecific antibody platform, called BiS4, against *P. aeruginosa* to significantly enhance protection and obtain broader coverage. They first identified two mAbs which independently targeted two antigens (exopolysaccharide PsI and type III secretion system virulence factor PcrV). They suggested combining two functionalities in one mAb, coupling anti-cytotoxic activity provided by anti-PrcV with anti-PsI mediated opsono-phagocytic and anti-adherence activities. The antibody showed activity against multiple strains, including multi-drug resistant ones, and in multiple infection models in prophylactic and therapeutic regimens. In fact, it resulted in a clinical candidate called MEDI3902, under development for the prevention or treatment of *P. aeruginosa* infections [56].

Given the low abundance of some antigens on the membrane, through bispecific mAbs, one may exploit the high abundance of one antigen to compensate for the reduced abundance of the other, which is less likely to be bound by an individual mAb. In the case of BiS4, the high-avidity binding activity against the highly abundant PsI was enough to enhance the activity against less-abundant PcrV, resulting in higher local mAb concentration. Another factor which can limit bispecific antibodies is the distance between the two targets on the bacterial surface, which, if distant, may make simultaneous binding of the bispecific mAb less likely to take place. To overcome this limit and allow simultaneous binding to both PsI and PrcV, the antibody was modified so that each binding unit is separated by a suitable intramolecular distance, allowing greater flexibility [56]. However, bispecific mAbs may not always provide the best solution. Despite proving that combining mAbs that target multiple virulence factors gives better results over each single mAb, when Tkaczyk et al. constructed bispecific mAbs, targeting ClfA and alpha-toxin, and tested them in vivo, they reported loss of protective activity compared to administration of single antibodies. In this case, the sequestration of the bispecific mAb to the bacterial surface, through high-affinity binding to ClfA, reduces its capacity to bind and neutralize the soluble alpha-toxin. Therefore, the location of the antigens, whether they are both surface-bound or soluble, may dictate the choice between mAb combination or bispecific mAbs [57]. In any case, these results suggest that binding multiple antigens may provide an approach superior to individual antigen targeting for inhibiting complex bacterial pathologies.

### 4.3. Fc Glyco-Engineering

Fc effector functions of antibodies are regulated by two processes: through Fc class-switch recombination, which spans different isotypes (i.e., IgG, IgM, IgA, IgD, and IgE) and through post-translational glycosylation [64]. Glycosylation is a chemical modification which consists in the addition of oligosaccharides (fucose, galactose, and sialic acid) to a conserved Asn297 N-glycosylation site in the heavy chain Fc region [72] Glycosylation has been found to regulate antibody stability, half-life, and immunogenicity, and it is thus critically important [73]. As the glycosylation of the Fc contributes to the conformation and structural integrity of the antibody, its biological activity is largely influenced as well. The presence of the glycan maintains the Fc in an open conformation, allowing the interaction between the IgG Fc region with the Fcγ receptor [74]. For example, fucose removal is a modification which increases the affinity of the antibody to FcγRIII, leading to an improved receptor-mediated effector function [75]. Agalactosylated Abs are instead associated with inflammation [76]. Furthermore, the glycosylation pattern variations of mAbs could protect mAbs from proteases. This could be due to the fact that, other than inducing conformational changes, the sugar moiety reduces the flexibility of the mAb and protects the cleavage sites from the proteolytic cleavage of enzymes [72].

Variation in fucosylation, galactosylation, and sialylation of the N-linked glycan gives the IgG Fc region a great level of heterogeneity. In fact, antibody glycosylation plays a major role in the response to viral or bacterial infection in humans. The immune system and inflammatory response can shape the glycosylation pattern of the antibody. In fact, individuals that better control bacterial infections have unique Fc-glycan profiles [77]. For example, in individuals infected by *Mycobacterium tuberculosis* (Mtb), it has been shown that the Fc-mediated antibody effector functions, tuned via differential glycosylation, can influence the ability of the immune system to control the infection. The outcome of the infection as latent tuberculosis infection (Ltb) or active tuberculosis disease (Atb) is associated with unique antibody Fc functional profiles and distinct antibody glycosylation patterns [78] IgG from individuals with Ltb compared to Atb contained less fucose, thus the Abs had an enhanced binding to FcγRIII and had fewer “inflammatory” agalactosylated antibodies [78].

As the efficacy of mAbs can be directly influenced by variations in the carbohydrate residues, Chen et al. evaluated whether glycosylation improved the efficacy of an anti-*Staphylococcus aureus* human mAb. The antibody binds and neutralizes the abundant surface-exposed Staphylococcal protein A (SpA). When produced by CHO cells, it did not display any therapeutic activity in a mouse model of MRSA infection. The defect correlated with low abundance of galactosylated antibodies in CHO cells. Enzymatic addition of galactosyl residues favored C1q recruitment, which allowed mAbs to exert opsonophagocytic activity against staphylococci. This, in turn, translated into protection against bloodstream infection in animals [79]. Furthermore, the same authors found that it is possible to tune mAbs towards a biased binding for C1q or FcγRs. Fucosylation favors interactions with C1q and inhibitory FcγRs while weakening interactions with activating FcγRs. If fucosylation of N-glycans is prevented, the mAbs may also engage FcγRs [79].

## 5. Future Perspectives

Compared to commonly used antibiotics, mAbs present clear advantages. However, a main aspect to take into consideration is how advantageous mAbs can be in terms of relative risk and cost compared to other treatment options. It is crucial to understand whether, in addition to manufacturing and control costs, the engineering procedures have a dramatic impact on the overall budget. Furthermore, more effort should be invested in reliable infectious models that mimic the disease patterns and the host target site. This would greatly improve the translation of the therapeutic mAb from the bench to the clinic and reduce the rate of clinical trial failure. This is crucial particularly for new mAb scaffold and structures, as those outlined in this review whose pharmacology had not been well-characterized previously, such that their potential adverse events could be less predictable [27].

## Figures and Tables

**Figure 1 biomedicines-10-02126-f001:**
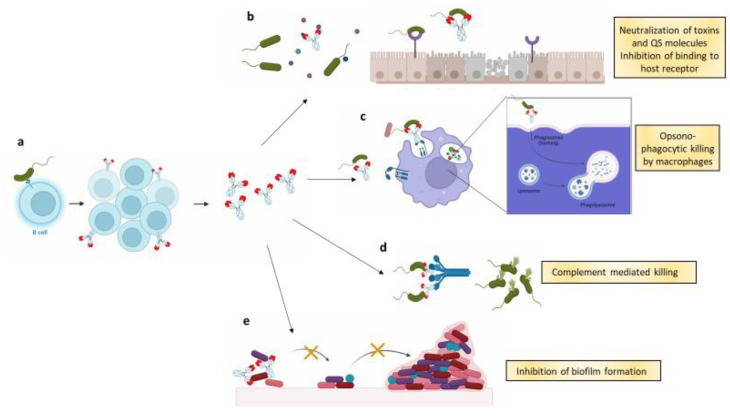
(**a**) B cells that are exposed to a single pathogen start replicating once they bind to a single virulence factor. The clones start expressing antibodies (monoclonal antibodies), which recognize the same virulence factor; (**b**) mAbs may act through Fab mediated functions, such as neutralization and inhibition of adhesion. In the first case, mAbs neutralize by binding to the toxins or to quorum sensing (QS) molecules released by the pathogenic bacteria, blocking the virulent effect exerted by the molecules. In the second case, mAbs bind to bacteria and inhibit their interaction with host cell’s receptors. In this way, bacteria cannot colonize the cell surface; (**c**) mAbs may enhance the opsono-phagocytic activity of macrophages. In some cases, they may also act intracellularly, promoting phago-lysosome formation and elimination of internalized bacteria; (**d**) once they bind to the pathogen, mAbs recruit complement components to initiate the complement cascade, resulting in bacterial lysis by the membrane attack complex (MAC); (**e**) lastly, mAbs that bind adhesins on bacterial surface disrupt the interactions within bacteria and between bacteria and the abiotic surface, thus interfering with biofilm formation.

**Table 1 biomedicines-10-02126-t001:** Mechanisms of Fc engineering for therapeutic IgG1 mAbs.

Fc Engineering	Mechanism	Reference
Fc engineering for enhanced effector engagement	Selective engagement of particular classes of human FcγRs	[15]
	Fc mutation leading to hexamerization upon antigen binding. This leads to greater engagement of Fc to C1q and C3b	[57,58]
	Fc portion multiplication, leading to incremental binding to FcgR, FcRn, and C1q through incremented avidity effect	[60]
Fc engineering for increased valency	Antibody carries two different Fabs, each specific for one antigen	[47]
Fc glycol-engineering	Addition of oligosaccharides to a conserved Asn297 N-glycosylation site in the heavy chain Fc region	[64]

## Data Availability

Not Applicable.

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
