# Peer review of "Monoclonal Antibodies for Bacterial Pathogens: Mechanisms of Action and Engineering Approaches for Enhanced Effector Functions"

_biomedicines, 2022, doi:10.3390/biomedicines10092126_

Round 1
Reviewer 1 Report
The review by Vacca and coauthors describe the application of monoclonal antibodies (mAbs) for treating bacterial infections and approaches for enhancing their neutralizing activity. This compact review is written interestingly, logically and can be very educational for bioscience students. However, the review can be significantly improved by addressing the following points:
1) The first sentence of the introduction is a bit confusing:
Line 30-32 “Monoclonal antibodies (mAbs) represent the predominant treatment option for various diseases [1] and are now the most important type of biological drugs on the pharmaceutical market [2].” – indeed the application of mAbs is considered to be very promising, but nowadays only ~100 mAbs have obtained approval by the FDA (line 43). Thus, this sentence should be reformulated to be less emotionally charged.
2) In line 83 the authors noted that the effector mechanism of antibodies is genetically different for humans and animals. Indeed, the mechanism of action of human IgG-1-subtype antibody (most promising for therapeutic use) is closer to another subtype (IgG-2a) in mice and the authors absolutely correctly noted that a great problem for the development of new antibacterial mAbs is translating results from animals to humans. I suppose that this review will significantly benefit if the authors briefly compare the effector action of antibodies in humans and animals and formulate their vision about the animal experiments which are necessary for establishing effectiveness of mAbs in the preclinical trials.
3) This review is mostly “Fc-focused”, however it will benefit from additional information about the specificity of antibacterial antibodies currently under development. Authors may summarize the information about target antigens within bacterial pathogens and the corresponding infection models. It would be convenient to present this information in the form of a table as, for example, in the review J. Fungi 2020, 6, 22; doi:10.3390/jof6010022.
4) Table 1 is confusing. I recommend to reformat it. It is better to present references as numbers in square brackets. The table column “Effector function enhanced” contains repetitive and excessive information and can be removed. The table column “Aim” would be better formulated as “Fc engineering” with the following parts “Fc engineering for enhanced receptor engagement”, “Fc engineering for increasing valency”, “Fc glyco-engineering”. Please check if the reference “Klimpel, K.R. et al. Proceedings of the National Academy of Sciences 1992, 98,10277-10281.” is correct. The phrase “Enhanced affinity of Fc to FcγR” needs to be specified.
The manuscript can be published in Biomedicines after revisions.
Reviewer 2 Report
A review entitled "Monoclonal antibodies for bacterial pathogens: mechanisms of action and engineering approaches for enhanced effector functions" is aimed to overview the mechanisms of action of mAbs against bacteria and recent advances in mAbs engineering to increase efficiency of anti-bacterial mAbs. The first part of the article explains the modes of action of the mAbs. It is well-structured and major information is presented and based on the appropriate citations. It really introduces the reader to the topic which is reviewed. In the second part the Fc engineering of the mAbs is reviewed. All the information explained is up-to-date and based on the citations. However, some future perspectives paragraph would enrich the review article. In my opinion, it should also include the information about the clincal performance of the three mAbs already approved by the FDA in order to prove the point that such pharmaceuticals are effective in human use. Nonetheless, the whole review article is well-written and comprehensible. It would contribute to the scientific discussion about the topic analyzed. I suggest to accept the article for publication after minor revision.
I have several specific comments for the general improvement of the review:
1. lines 119-124. In this paragraph some additional information would enrich the review. In contrary to the text, the inflammation activated through TLR4 mobilize innate immune system to fight and is very important to eradicate bacteria. Is there any research about the interplay of normal immune response and therapeutic mAbs?
2. line 135. I suggest to delete word "also" as it may mislead a reader.
3. line 160. "ADCC" mentioned for the first time, so it should be explained.
4. line 162. "CDC" mentioned for the second time, so the abbreviation only should be used.
5. line 262. "e" should be changed to "and".
Round 2
Reviewer 1 Report
In my opinion, the review was significantly improved during revision.
About specificity of antibacterial antibodies, I agree with authors that discussion about specificity and listing all the antigens of the antibodies currently under development may take us far away from the subject. But it may be good idea for further “Fab-focused” review?